# Rethinking Data Selection at Scale: Random Selection is Almost All You Need

## Abstract

Supervised fine-tuning (SFT) is crucial for aligning Large Language Models (LLMs) with human instructions. The primary goal during SFT is to select a small yet representative subset of training data from the larger pool, such that fine-tuning with this subset achieves results comparable to or even exceeding those obtained using the entire dataset. However, most existing data selection techniques are designed for small-scale data pools, which fail to meet the demands of real-world SFT scenarios. In this paper, we replicated several self-scoring methods—those that do not rely on external model assistance—on two million-scale datasets, and found that nearly all methods struggled to significantly outperform random selection when dealing with such large-scale data pools. Moreover, our comparisons suggest that, during SFT, diversity in data selection is more critical than simply focusing on high-quality data. We also analyzed the limitations of several current approaches, explaining why they perform poorly on large-scale datasets and why they are unsuitable for such contexts. Finally, we found that filtering data by token length offers a stable and efficient method for improving results. This approach, particularly when training on long-text data, proves highly beneficial for relatively weaker base models, such as Llama3.

## 1 Introduction

With the advent of large language models (LLMs) such as ChatGPT, we have observed significant advancements in tasks involving instruction following (Wang et al., 2023b), intent comprehension (Lu et al., 2023), and text generation (Zhao et al., 2023). One of the primary objectives of developing LLMs is to harness their potential for generalizing to unseen natural language processing (NLP) tasks. To achieve this aim, many LLMs focus on precisely aligning with human instructions.

Recent studies indicate that supervised fine-tuning (SFT) can customize LLMs for specific domains, tasks, or applications by utilizing well-crafted data. According to the study in Zhou et al. (2024a), it is feasible to fine-tune a pre-trained language model with a relatively small set of examples. Building on this insight, several papers have explored data selection strategies for SFT of LLMs (Wang et al., 2024; Qin et al., 2024), emphasizing the importance of enhancing the quality of instruction tuning (IT) data or increasing data diversity. These strategies can be classified into two primary categories: (1) Extenral-scoring methods, which require support from more sophisticated external models like GPT-4 to score the data for the subsequent selection (Lu et al., 2023; Chen et al., 2023; Du et al., 2023; Liu et al., 2023; Zhou et al., 2024b); (2) Self-scoring methods, which leverage LLMs themselves as data scorers (Zhou et al., 2023a; Li et al., 2023d;b; Liu et al., 2024; Xia et al., 2024; Yin et al., 2024).

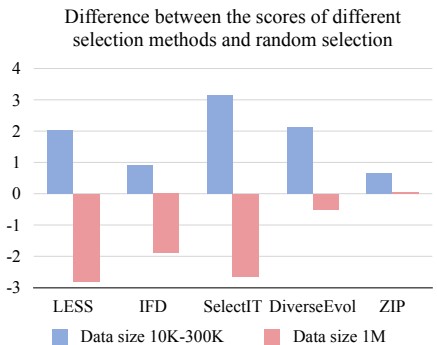

Difference between the scores of different selection methods and random selection

Figure 1: The discrepancy between each methods and random selection on BBH benchmark (Suzgun et al., 2022). The Y-axis represents the differential score, which is computed by subtracting the random selection score from the scores obtained using various methods.

Existing SFT data selection methodologies, both external-scoring and self-scoring, are primarily assessed using several widely recognized IT datasets, such as alpaca-GPT4 (Peng et al., 2023), Dolly (Conover et al., 2023), FLAN (Longpre et al., 2023), WizardLM (Xu et al., 2024), and ShareGPT (Chiang et al., 2023). These datasets are limited in size and originate from a single source. However, during the SFT stage, a substantially larger scale of data, typically ranging from hundreds of thousands to even millions in size, is frequently necessary. For example, Qwen2 (qwe, 2024) utilized over 500,000 pieces of data during the SFT process. Therefore, in practical applications, in order to fully utilize the inherent knowledge of LLMs, large-scale instruction-following data is essential in the SFT process. Moreover, large-scale data sources not only require a sufficient amount of data, but should also have diverse data sources, such as annotated by professional workers, sourced from real users, or synthesized by models, and rich data types include code data, math data, conversation content, knowledge Q&A, etc.. This discrepancy creates a gap between the present SFT data selection strategies and real-world applications. In order to observe the impact brought by the dataset size on the performance of different selection strategies, we analyze the difference in outcomes between existing SFT data selection methods and random selection within source datasets ranging from 10K-30K to 1M on Llama3-8B (AI@Meta, 2024). As shown in Figure 1, when the scale of the datasets increases to 1M, these data selection methods yield suboptimal performance compared with random selection. Here, "Data size 10K-300K" refers to the data sources used in the original papers of different methods. "Data size 1M" refers to Openhermes2.5-1M dataset (Teknium, 2023).

Inspired by this finding, we rethink whether SFT data selection methods can work when they are required to handle large-scale IT datasets. For external-scoring approaches, it is impractical to apply them to tackle vast amounts of IT data due to the substantial costs (Liu et al., 2023), we hence focus on the self-scoring methods. For self-scoring approaches, we refer to the article Qin et al. (2024) to categorize the techniques into two types: data quality-based methods and data diversity-based methods. Data quality-based methods imply that the approach lays greater emphasis on devising an algorithm and evaluation metrics to compute the score of each data item. Subsequently, the selection is carried out based on the data scores. In contrast, the data diversity-based method is more centered around the diversity of the dataset. To explore how self-scoring methods influence LLMs' performance when dealing with large-scale IT data, we evaluate several recent methods on two benchmarks that contain millions of instances. The findings from our experiments reveal three main points:

- Most self-scoring data selection techniques do not significantly outperform random selection on large-scale datasets. Even though these self-scoring methods can achieve significant gains on small-scale datasets, their effectiveness will be greatly reduced when the data size increases and the data sources become complex. While the performance of certain methods does exhibit a marginal edge over the random approach when implemented on particular LLMs, a comprehensive consideration of the trade-off between effectiveness and efficiency leads us to the conclusion that, when dealing with extensive data sources, random selection stands out as the most preferable and advantageous option.

- Data diversity holds more significance than data quality during the SFT phase. Data quality-based selection methods are more effective than data diversity-based methods when dealing with a small-scale dataset from a single source. However, when tackling multi-source data, only considering data quality is far from enough.

- Through a comparative empirical analysis of two IT datasets, we find that it is useful to utilize token length as a criterion to conduct data filtering, yielding stable and efficient results for SFT when dealing with large-scale IT data. Previous work (Liu et al., 2023) has demonstrated the benefit of long texts training for models on subjective evaluation tasks such as MTbench (Zheng et al., 2023) and AlpacaEval (Li et al., 2023c), we have further confirmed the positive effect of long texts training on objective evaluation tasks, such as Big-Bench-Hard (Suzgun et al., 2022). While utilizing token length in SFT may not yield optimal outcomes on every language model, it is highly beneficial for applying it in training with long texts, especially on a relatively weak BASE language model, like Llama3-8B.

## 2 RELATED WORK

**External-scoring Method**. Lu et al. (2023) introduced an open-set instruction tagging method called INSTAG, which employed ChatGPT to generate detailed tags to measure and examine the variety and intricacy of human instructions for LLMs during SFT. Chen et al. (2023) presented the ALPAGASUS model that used ChatGPT to evaluate each instruction and then selected various data based on a certain threshold. Du et al. (2023) suggested a model-oriented instruction selection approach that not only considered the quality and coverage of instruction data but also incorporated the necessity of instructions according to the capabilities of specific LLMs. Liu et al. (2023) introduced DEITA, it used ChatGPT to iteratively enhance the complexity or quality of each data sample across relevant dimensions and then requested ChatGPT to evaluate these samples for their complexity or quality. These models exceed the performance of the basic foundation models trained on complete datasets. However, they heavily depend on high-performing external LLMs to score data.

**Self-scoring Method**. Li et al. (2023b) put forward an autonomously guided method enabling LLMs to discern relevant instruction pairs from open-source data. An Instruction-Following Difficulty (IFD) metric was introduced to highlight inconsistencies between a language model's anticipated responses and its self-generated outputs. Wu et al. (2023) came up with DiverseEvol, which enabled the model to progressively select training subsets to enhance performance, without external oversight from humans or more advanced LLMs. This approach focused on maintaining high diversity within the selected subsets, as the model opted for new data points that are most distinct from existing ones based on its current embedding space. Xia et al. (2024) suggested LESS, designed to pick out relevant instruction tuning data for a specific application. It utilized a gradient datastore with low-dimensional gradient features, selecting examples based on their resemblance to few-shot examples that represent a particular capability. Yin et al. (2024) observed that model performance is inversely related to the compression ratio of training data. They introduced a universal data selection method named ZIP aimed at prioritizing data subsets with low compression ratios for training LLMs. Liu et al. (2024) developed SelectIT, which leveraged the inherent uncertainty in LLMs at various levels—grain, token, sentence, and model—to more effectively identify high-quality instruction tuning data, eliminating the need for additional resources. Li et al. (2023d) introduced Nuggets, which employs one-shot learning to choose high-quality instruction data. It used a scoring system based on the influence of candidate examples on the perplexity of a diverse anchor set, thereby facilitating the selection of the most beneficial data for instruction tuning.

## 3 SELF-SCORING STRATEGIES

In this paper, we focus on self-scoring methods that do not rely on external advanced LLMs to score data. We refer Qin et al. (2024)'s work and categorize existing resourceful data selection methods into two main perspectives: data quality-based methods and data diversity-based methods.

### 3.1 QUALITY-BASED SELECTIONS

In this section, we introduce 4 methods based on data quality assessment and selection. "Quality" here refers primarily to the complexity, completeness, score, and influence of the datapoints. Different from Qin et al. (2024), we believe that the influence of a datapoint in the target dataset is also a reflection of data quality, especially in practical scenarios, where we are required to deal with diverse tasks rather than a single task. We thus regard the influence as a quality category as well.

**LESS** Xia et al. (2024) instroduced low-rank gradient similarity search to select influential data for the target application. Concretely, a model was trained with LoRA (Hu et al., 2021) for a warmup period on a small subset $\mathcal{D}_{\mathrm{warmup}} \subset \mathcal{D}$. Then, the Adam LoRA gradient features for each data point were computed and stored in a gradient database.

Next, a gradient datastore of projected low-dimensional gradient features was constructed which can be reused for different target tasks. For training datapoints $\boldsymbol{x}$, they computed d-dimensional projection of the LoRA gradient $\tilde{\nabla}\ell(\boldsymbol{x};\boldsymbol{\theta}_i) = \Pi^\top \hat{\nabla}\ell(\boldsymbol{x};\boldsymbol{\theta}_i)$, where $\Pi^\top$ is computed and applied by memory-efficient online implementation of random projections proposed by Park et al. (2023). For validation datapoint $\boldsymbol{x}'$, they computed $\tilde{\Gamma}(\boldsymbol{x}',\cdot) = \Pi^\top \hat{\Gamma}(\boldsymbol{x}',\cdot)$, where $\tilde{\Gamma}(\boldsymbol{x}',\cdot)$ represents the gradient values of different data $\boldsymbol{x}'$ under different optimization states $\cdot$.

Finally, LESS computed $\max_j \text{Inf}_{\text{Adam}}(\boldsymbol{x}, \mathcal{D}_{\text{val}}^{(j)})$ for the training set $\boldsymbol{x}$ across all sub-validation sets $\mathcal{D}_{val}$. Then it selected the highest score examples to construct $\mathcal{D}_{train}$.

$$\text{Inf}_{\text{Adam}}(\boldsymbol{x}, \mathcal{D}_{\text{val}}^{(j)}) = \sum_{i=1}^{N} \bar{\eta}_i \frac{\langle \bar{\nabla}\ell(\mathcal{D}_{\text{val}}^{(j)}; \boldsymbol{\theta}_i), \tilde{\Gamma}(\boldsymbol{x}, \boldsymbol{\theta}_i) \rangle}{\|\bar{\nabla}\ell(\mathcal{D}_{\text{val}}^{(j)}; \boldsymbol{\theta}_i)\| \|\tilde{\Gamma}(\boldsymbol{x}, \boldsymbol{\theta}_i)\|} \tag{1}$$

**IFD** introduced the Instruction-Following Difficulty (IFD) score, a metric devised to evaluate the challenge each instructional sample presents (Li et al., 2023b). Given a $(Q, A)$ pair, they calculated the ratio between $s(A)$ and $s(A|Q)$:

$$\text{IFD}(Q, A) = \frac{s(A|Q)}{s(A)} = \frac{-\frac{1}{N}\sum_{i=1}^{N}\log P(x_i^A|Q, x_1^A, x_2^A, \dots, x_{i-1}^A)}{-\frac{1}{N}\sum_{i=1}^{N}\log P(x_i^A|x_1^A, \dots, x_{i-1}^A)} \tag{2}$$

where $s(A)$ means Direct Answer Score, which measures LLM's ability to generate the answer alone. $s(A|Q)$ means Conditioned Answer Score, which is calculated by continuously predicting the next tokens given the instruction $Q$ and their proceeding words.

In this paper, the authors first generated 100 clusters on instruction embeddings and sampled 10 instances in each cluster based on IFD score on pre-trained base LLM. Then they trained that LLM for 1 epoch by using the selected datapoints. After training, they calculated the IFD score of each datapoint of the whole training set $\mathcal{D}$ and finally selected the highest IFD score data as $\mathcal{D}_{train}$.

**SelectIT** selected high-quality IT data based on the intrinsic uncertainty reflected by LLMs (Liu et al., 2024). It included three grains of sample evaluation modules: token, sentence, and model level self-reflections.

For token level, SelectIT calculated the probability of the next token (from 1 to $K$) based on the rating prompt $RP$ and query-response pair $E$. The score token with the highest probability was then considered as the quality of the sample. The higher $P'_{E^{base}}$, the more confidence of LLMs

$$E^{base} = \arg\max P'_k, P'_k = \left(\frac{e^{P_k}}{\sum_{j=1}^{K} e^{P_j}}\right) \tag{3}$$

where $P_k$ and $P'_k$ mean the probability and softmax probability of token $k$. K means the number of scores to be considered. In that paper, the score token ranged from 1 to 5. To enhance the credibility of quality assessment, SelectIT assessed the average disparity between the predicted token $E^{base}$ and the other, where the greater the disparity, the greater the confidence of the LLM.

$$E^{token} = E^{base} \times \frac{1}{K-1} \sum_{i=1}^{K} |P'_i - P'_{E^{base}}| \tag{4}$$

For sentence level, since different prompts can significantly affect outputs of LLMs, it designed $K$ semantically similar rating prompts $\{RP_0, RP_1, \dots, RP_K\}$ and obtained a series of quality scores $\{E_0^{token}, E_1^{token}, \dots, E_K^{token}\}$, respectively.

$$E^{sent} = \frac{\mathbf{Avg}\{E_i^{token}\}_{i=1}^{K}}{1 + \alpha \times \mathbf{Std}\{E_i^{token}\}_{i=1}^{K}} \tag{5}$$

where $\mathbf{Avg}\{\cdot\}$ and $\mathbf{Std}\{\cdot\}$ denote the mean and standard deviation of $E_i^{token}$, respectively. $K$ means the number of rating prompts $RP$.

For model level, SelectIT used $N$ foundation models with parameter counts $\{\beta_1, \beta_2, \dots, \beta_N\}$ and their respective sentence-level scores for a sample E being $\{E_0^{sent}, E_1^{sent}, \dots, E_N^{sent}\}$, then the model-level score $E_{model}$ was computed as follows.

$$E^{model} = \sum_{i=1}^{N} \left(\frac{\beta_i}{\sum_{j=1}^{N} \beta_j} \times E_i^{sent}\right) \tag{6}$$

where $N$ means the number of the foundation models. It used $E_{model}$ as the final evaluation of sample $E$ in SelectIT.

**Cross-entropy**: Language models can be considered a form of compression, with LLMs showing strong capabilities in data compression empirically (Delétang et al., 2024). Compression efficiency is a stable and reliable assessment that is linearly related to the model's capabilities. It reflects the model's ability to extract relevant information and eliminate unnecessary elements, providing insight into the intrinsic capability of the language model (Huang et al., 2024; Wei et al., 2024).

The cross-entropy loss employed in the training of LLMs establishes a coherent relationship between LLMs and information compression of each query-response pair $E$.

$$\mathbb{E}_{x^E \sim \rho}[-\sum_{i=1}^{n} \log_2 \rho_{model}(x_i^E | x_{1:i-1}^E)] \tag{7}$$

Inspired by this foundational insight, we select data based on the cross-entropy of each datapoint, where the higher value of cross-entropy means the better quality.

## 3.2 DIVERSITY-BASED SELECTIONS

In this section, we introduce methods that emphasize the diversity of instruction datasets, where diversity refers to the overall diversity of the entire training dataset.

**DiverseEvol** iteratively sampled training subsets to improve its own performance (Wu et al., 2023). It selected new data points most distinct from any existing ones according to its current embedding space in each iteration phase.

Given a training set $\mathcal{D}$, DiverseEvol first randomly selected a data pool $P_0$ and trained an initial model $M_0$. In each iteration, it consisted of two operations: 1. Deduce new data points $\mathcal{D}_t$ to merge into $P_{t+1}$, informed by the previously trained model $M_t$. 2. Train the subsequent chat model $M_{t+1}$, with the updated data pool $P_{t+1}$.

DiverseEvol used K-Center-Sampling to select data. From a candidate pool, it chose $k$ data points in such a way that the distances to their respective nearest existing training data points were maximized.

$$\arg\max_{i \in X_t} \min_{j \in P_t} \Delta\left(\boldsymbol{x_i}, \boldsymbol{p}_j\right) \tag{8}$$

At each step, the input parameters to K-Center-Sampling were the model $M_t$, the current training pool $P_t$, and $\mathcal{D}_t$. The selection function K-Center-Sampling then outputs the new data point $X_t$, which was added to the training pool for the next iteration $P_{t+1}$.

**ZIP** presented that model performance is negatively correlated to the compression ratio of training data, which usually yields a lower training loss. Yin et al. (2024) proposed a quite efficient and universal data selection method named ZIP for training LLMs, which aimed to prioritize data subsets exhibiting a low compression ratio.

ZIP is initialized by calculating the sample-level compression ratio for the entire dataset $\mathcal{D}$, where $\pi_{\mathcal{D}}$ shows the information redundancy state of $\mathcal{D}$. In each iteration, it selected $K_1$ samples with the lowest $\pi_{\mathcal{D}_1}$ to form an initial candidate pool $\mathcal{D}_{K_1}$. Then, it calculated the compression ratio of a merged set that adds each sample in $\mathcal{D}_{K_1}$ to the selected set $\mathcal{D}_{train}$, to update the redundancy state of the information $\pi_{\mathcal{D}_1}$.

Based on the scores of the samples in $\mathcal{D}_{K_1}$, ZIP selected $\mathcal{D}_{K_2}$ samples with the lowest scores. After that, it initialized an empty selected set $\mathcal{D}_{K_3}$, and computed the compression ratio of the union of $\mathcal{D}_{K_3}$ and each sample in $\mathcal{D}_{K_2}$. Then, the sample with the lowest compression ratio was added to $\mathcal{D}_{K_3}$, and removed from $\mathcal{D}_{K_2}$. Finally, each sample in $\mathcal{D}_{K_3}$ was added to the selected set $\mathcal{D}_{train}$. In ZIP, the compression ratio calculation $g(\mathcal{C}(D))$ is defined as:

$$g(\mathcal{C}(D)) = \frac{\text{Bits}(D)}{\text{Bits}(\mathcal{C}(D))} \tag{9}$$

where $\mathcal{C}$ means the compression ratio.

## 4 EXPERIMENT

### 4.1 DATASETS

In practical applications, researchers frequently encounter extensive datasets from various sources during SFT, which may also contain imperfections. Thus, in this study, rather than using the typically employed IT datasets such as alpaca (Taori et al., 2023), we select two large-scale IT datasets at the million-record level, Openhermes2.5 (Teknium, 2023) and WildChat-1M (Zhao et al., 2024), to examine the efficiency of existing data selection techniques in handling large datasets and to assess their performance in real-world scenarios.

**Openhermes2.5** is presented by Teknium (2023), which comprises over 1 million data points. It is significantly more comprehensive and of higher quality, predominantly consisting of generated guides and chats. The dataset's information is sourced from 16 distinct origins, including meta-math (Yu et al., 2023), CamelAI (Li et al., 2023a), among others. It encompasses a wide variety of subjects such as mathematics, programming, and authentic user dialogues.

**WildChat-1M** is introduced by Zhao et al. (2024) and features solely non-toxic user inputs and ChatGPT responses. The dataset comprises 1 million dialogues between human users and ChatGPT, with 25.53% of the interactions stemming from the GPT-4 model, and the remainder from GPT-3.5. It encompasses a diverse range of user-chatbot exchanges, including ambiguous user inquiries, code-switching, topic-switching, and political discussions. In this study, we extract English dialogues from the WildChat dataset, resulting in over 440k interactions.

### 4.2 BENCHMARKS

To thoroughly evaluate the capabilities of LLM, we explored various approaches across different downstream tasks. We assess the reasoning abilities of LLMs using two commonly used datasets: the Grade School Math dataset (GSM) (Cobbe et al., 2021) and Big-Bench-Hard (BBH) (Suzgun et al., 2022) within the CoT setting (Wei et al., 2022). We evaluate the code generation capability with the HumanEval dataset (Chen et al., 2021) and report pass@1 results. To determine the factual knowledge of LLMs, we use the Massive Multitask Language Understanding dataset (MMLU) (Hendrycks et al., 2021) and provide 5-shot results. We also assess instruction-following ability using the IFEval (Zhou et al., 2023b) dataset and report both strictly and loosely followed scores. Additionally, we utilize scripts from OpenInstruct, which includes a collection of standard benchmarks focusing on core capabilities (Wang et al., 2023a; Ivison et al., 2023; 2024).

### 4.3 IMPLEMENTATION DETAILS

Specifically, we leverage the widely-used LLaMA3-8B (AI@Meta, 2024) and Qwen2-7B (qwe, 2024) as our base models, and fine-tune them using the Llama-Factory framework (Zheng et al., 2024). We train these models for 3 epochs with a batch size of 128. Our training process employs a cosine learning rate scheduler beginning at $7e - 6$, which decays to 0.1, warms to 0.01, and utilizes an input length of 4096. To replicate our baseline methods on Openhermes and WildChat, we adjust some original parameters and implementations to fit the large-scale datasets.

In term of LESS, individual models are built and trained on specific tasks. However, in practical applications, our goal is to train a model that enhances performance across various scenarios. Thus, given that the two datasets we select are both extensive and diverse, we randomly select 1000 data points from each dataset as $\mathcal{D}_{val}$. Additionally, due to the volume of our data, we randomly pick 10,000 data points for warm-up training, differing from the method described in (Xia et al., 2024).

As for IFD, we initially generate 1000 clusters on instruction embeddings, which differs from the settings given in Li et al. (2023b). For SelectIT, we adopt model-level selection as the final strategy for the Qwen2 model and evaluate the model-level score on Qwen2-1.5B and Qwen2-7B. While for Llama3, we employ sentence-level selection as the final approach. Considering that the Llama3 family only has two public variants, Llama3-8B and Llama3-70B, and to mitigate time costs, we compute the score based solely on Llama3-8B.

Within DiverseEvol, during each iteration's K-Center-Sampling stage, data points are selected based on maximizing their distance to the nearest existing training data points, one at a time, until the

Table 1: The overall results (%) on a variety of downstream tasks based on Openhermes2.5 dataset. CODE means HumanEval, Random $n$ denotes the $n$th random selection. Except for fine-tuning with the entire Openhermes dataset, the bold numbers indicate the best score of each part, and the underlined numbers indicate the second highest score.

| | Qwen2-7B | | | | | | | Llama3-8B | | | | | | |
| | BBH | GSM | CODE | MMLU | IFEVAL | | AVG | BBH | GSM | CODE | MMLU | IFEVAL | | AVG |
| | 3 shot | 8 shot | pass 1 | 5 shot | strict | loose | | 3 shot | 8 shot | pass 1 | 5 shot | strict | loose | |
| Base | 59.07 | 72.40 | 55.67 | 70.20 | 28.84 | 31.24 | 52.90 | 60.93 | 55.12 | 37.59 | 65.30 | 19.41 | 21.07 | 43.24 |
| all data | 61.39 | 80.12 | 63.32 | 68.50 | 40.85 | 44.18 | 59.73 | 63.33 | 73.24 | 46.43 | 63.90 | 46.40 | 49.72 | 57.17 |
| Random 1 | 59.72 | 82.41 | 62.10 | 68.30 | 33.27 | 36.41 | 57.04 | _64.72_ | 53.90 | 45.21 | 63.20 | 39.19 | 43.62 | 51.64 |
| Random 2 | _61.48_ | **83.47** | 64.33 | 67.90 | _38.08_ | _40.30_ | _59.26_ | 60.83 | 56.86 | **48.99** | 62.70 | 41.77 | 45.47 | 52.77 |
| Random 3 | **61.85** | 81.65 | 62.90 | 68.10 | 36.78 | 38.45 | 58.29 | 63.43 | _59.74_ | _46.83_ | 62.70 | 43.25 | 46.21 | _53.69_ |
| Random 4 | 61.20 | _82.71_ | 59.27 | 68.00 | 36.60 | 39.19 | 57.83 | 63.98 | 59.59 | 45.18 | 63.80 | _44.36_ | _47.13_ | **54.01** |
| Random 5 | 61.30 | _82.71_ | 62.23 | **68.90** | 35.86 | 37.71 | 58.12 | 62.31 | 56.10 | 42.07 | 63.50 | **44.55** | **48.80** | 52.89 |
| LESS | 61.20 | 81.65 | 53.26 | 67.60 | 32.16 | 37.15 | 55.50 | 61.39 | 57.70 | 41.43 | **64.20** | 38.08 | 41.96 | 50.79 |
| IFD | 57.96 | 79.23 | **68.48** | 56.70 | 33.27 | 35.12 | 55.13 | 57.41 | 53.53 | 32.41 | 59.90 | 43.07 | 45.84 | 48.69 |
| SelectIT | 59.17 | 80.44 | _66.46_ | 67.20 | 35.86 | 38.82 | 57.99 | 62.59 | **61.56** | 42.38 | 63.60 | 38.45 | 42.14 | 51.79 |
| Entropy | 61.30 | 55.04 | 61.04 | **68.90** | 37.34 | 40.48 | 54.02 | 58.61 | 50.72 | 44.02 | 61.40 | 32.90 | 37.89 | 47.59 |
| Diverse | 61.11 | 81.73 | 61.71 | _68.65_ | **40.85** | **43.44** | **59.58** | **65.00** | 56.25 | 44.51 | _63.84_ | 43.99 | 47.13 | 53.45 |
| ZIP | 60.65 | 80.52 | 66.10 | 68.60 | 37.15 | 39.56 | 58.76 | 63.98 | 59.67 | 40.70 | 62.60 | 43.81 | 46.58 | 52.89 |

desired count is reached. Consequently, it is essential to maintain a $n \times n$ float-type matrix for the entire computation, where $n$ represents the dataset size. Given that our OpenHermes dataset exceeds 1 million entries, the matrix calculation would require more than 1 terabyte of memory. Therefore, we revised this part to select all required data points once for each iteration, which significantly reduces the memory requirement.

## 5 DISCUSSION

### 5.1 BASELINE METHODS VS RANDOM

In this section, we reproduce all baseline methods in experiments involving LLaMA3-8B and Qwen2-7B on OpenHermes2.5, the experimental results are presented in Table 1, and results on WildChat are detailed in Table 3. We assess LLaMA3-8B and Qwen2-7B with and without fine-tuning on the entire dataset. All mentioned SFT data selection methods are employed to select 10,000 samples as described in Section 4.3. We randomly run 5 times and all of the results are provided in the tables. Furthermore, 50,000 samples obtained through various methods are also shown in the Appendix Table 6, 7.

As indicated in Table 1 and 3, it is evident that when dealing with extensive and diverse IT datasets, no data selection techniques consistently outperform random sampling by a substantial margin, which implies that the average score exceeds the random score by more than 1%. In most cases, the results of the baseline method are within the range of the results obtained by 5 random runs, and a few methods are even worse than the worst random result,

Table 2: The P-values of the significance tests for each method against the results of five rounds of random selection.

| | Llama3-8B | | Qwen2-7B | |
| | OpenHermes | WildChat | OpenHermes | WildChat |
| LESS | 0.77 | 0.45 | 0.80 | 0.86 |
| IFD | 0.85 | 0.53 | 0.85 | 0.68 |
| SelectIT | 0.71 | 0.79 | 0.60 | 0.58 |
| Entropy | 0.92 | 0.46 | 0.78 | 0.30 |
| Diverse | 0.39 | 0.58 | 0.37 | 0.45 |
| zip | 0.55 | 0.36 | 0.42 | 0.31 |

For instance, when evaluating Cross-Entropy on Qwen2-7B using Openhermes2.5, the average result is a mere 54.02, significantly below the lowest score of 57.04 obtained in the 5 random trials. Besides, We also conducted the Mann-Whitney U test for each method against the results of 5 rounds of random selection. We adopted the right-tailed test approach, with the testing hypothesis being that the scores of each baseline method on different test tasks are greater than those of the random method. We reported the p-value for each method being significantly better than that of the random

Table 3: The overall results (%) on a variety of downstream tasks based on WildChat dataset. CODE means HumanEval, Random $n$ denotes the $n$th random selection. Except for fine-tuning with the entire Openhermes dataset, the bold numbers indicate the best score of each part, and the underlined numbers indicate the second highest score.

| | Qwen2-7B | | | | | | | Llama3-8B | | | | | | |
|---|---|---|---|---|---|---|---|---|---|---|---|---|---|---|
| | BBH | GSM | CODE | MMLU | IFEVAL | | AVG | BBH | GSM | CODE | MMLU | IFEVAL | | AVG |
| | 3 shot | 8 shot | pass 1 | 5 shot | strict | loose | | 3 shot | 8 shot | pass 1 | 5 shot | strict | loose | |
| Base | 59.07 | 72.40 | 55.67 | 70.20 | 28.84 | 31.24 | 52.90 | 60.93 | 55.12 | 37.59 | 65.30 | 19.41 | 21.07 | 43.24 |
| all data | 62.87 | 80.82 | 62.84 | 68.70 | 45.84 | 48.80 | 61.65 | 63.70 | 56.94 | 47.44 | 63.30 | 46.40 | 49.72 | 54.58 |
| Random 1 | 61.30 | **82.64** | 61.98 | 68.10 | 40.30 | 42.33 | 59.44 | 63.70 | 56.48 | 51.92 | 63.30 | 39.37 | 41.95 | 52.79 |
| Random 2 | 60.93 | 81.96 | 61.43 | 67.50 | 38.63 | 40.67 | 58.52 | 62.41 | 52.62 | **49.33** | 64.00 | 44.18 | 46.77 | 53.22 |
| Random 3 | 60.28 | **82.64** | 62.07 | 68.30 | 41.04 | 42.88 | 59.54 | 63.52 | 58.38 | 43.90 | 64.10 | 42.33 | 45.29 | 52.92 |
| Random 4 | 61.11 | 80.36 | 65.46 | 67.50 | 37.34 | 40.67 | 58.74 | 63.33 | 55.42 | 51.10 | 64.50 | 41.96 | 44.55 | 53.48 |
| Random 5 | 61.57 | 81.50 | 60.27 | 68.20 | 41.77 | 43.99 | 59.55 | **64.91** | 60.27 | 48.66 | 64.30 | 42.14 | 45.84 | 54.35 |
| LESS | 52.59 | 60.50 | 61.19 | 68.00 | 38.82 | 41.77 | 53.81 | 63.43 | 57.01 | 50.43 | 64.50 | 40.85 | 44.92 | 53.52 |
| IFD | 60.56 | 76.27 | 65.24 | 68.00 | 36.23 | 38.26 | 57.43 | 63.33 | 59.29 | 47.16 | **64.60** | 40.30 | 43.81 | 53.08 |
| SelectIT | 60.37 | 82.34 | 64.97 | **68.50** | 36.97 | 39.19 | 58.72 | 61.48 | 53.22 | 46.01 | 63.20 | 40.11 | 42.88 | 51.15 |
| Entropy | 60.37 | 81.96 | 62.90 | 68.40 | **42.51** | **46.21** | 60.39 | 63.15 | 56.10 | 47.71 | 63.00 | 45.10 | 49.54 | 54.10 |
| Diverse | 61.02 | 80.82 | 65.09 | 67.33 | 41.04 | 42.88 | 59.70 | 62.59 | 53.30 | 33.48 | 64.46 | **47.87** | **50.65** | 52.06 |
| ZIP | **62.59** | 81.80 | **68.17** | 68.00 | 40.11 | 42.33 | **60.50** | 62.31 | **60.96** | 46.58 | 64.50 | 45.10 | 48.06 | **54.59** |

Table 4: The overall results (%) of token length selection.

| | Qwen2-7B | | | | | | | Llama3-8B | | | | | | |
|---|---|---|---|---|---|---|---|---|---|---|---|---|---|---|
| | BBH | GSM | CODE | MMLU | IFEVAL | | AVG | BBH | GSM | CODE | MMLU | IFEVAL | | AVG |
| | 3 shot | 8 shot | pass 1 | 5 shot | strict | loose | | 3 shot | 8 shot | pass 1 | 5 shot | strict | loose | |
| OpenHermes | 60.65 | 80.74 | 60.18 | 68.33 | 37.89 | 41.40 | 58.20 | 64.63 | 61.33 | 45.70 | 64.41 | 48.43 | 52.87 | 56.23 |
| WildChat | 61.67 | 81.05 | 59.21 | 67.82 | 39.56 | 42.14 | 58.58 | 66.11 | 60.35 | 51.16 | 63.91 | 43.81 | 47.69 | 55.51 |

method in table 2. We found that the p-values of all methods is higher than 0.05, which indicates that the results of all baseline methods are not greater than those of the random method.

Based on the experimental results, **when dealing with an extensive SFT dataset, it is more efficient to randomly select training data instead of spending significant time and resources to meticulously choose seemingly optimal training data.** Random selection reduces costs and yields superior training results.

## 5.2 QUALITY VS DIVERSITY

Tables 1 and 3 demonstrate that the diversity-based selection strategy outperforms the quality-based one. To examine whether prioritizing diversity over data quality improves data selection, we designed a supplementary experiment by incorporating a K-means clustering process on the OpenHermes dataset. Instead of selecting data based solely on method scores, we choose higher-scoring data within each cluster to boost the final training set's diversity.

Table 5 illustrates that integrating the K-means clustering with quality-based selection methods enhances the effectiveness for most approaches. Notably, Cross Entropy on both Llama3 and Qwen2 models shows improvement over 5% and 3%, respectively, when K-means is used to diversify the data. This suggests that for a large-scale IT dataset, **data diversity holds more importance than data quality**. This also clarifies why random selection often outperforms most SFT data selection methods, as the random process preserves the dataset's original distribution and diversity to the greatest possible extent.

## 5.3 BASELINE ANALYSIS

In this part, we mainly analyze several methods and try to find the reasons why these methods fail in large-scale data sets and why these methods are not applicable to practical applications.

Table 5: The overall results (%) on a variety of downstream tasks based on Openhermes2.5 dataset. Method$_{km}$ means method with kmeans process. The bold number indicates the avg performance increase after add K-means phase.

| | Qwen2-7B | | | | | | | Llama3-8B | | | | | | |
|---|---|---|---|---|---|---|---|---|---|---|---|---|---|---|
| | BBH | GSM | CODE | MMLU | IFEVAL | | AVG | BBH | GSM | CODE | MMLU | IFEVAL | | AVG |
| | 3 shot | 8 shot | pass 1 | 5 shot | strict | loose | | 3 shot | 8 shot | pass 1 | 5 shot | strict | loose | |
| LESS | 61.20 | 81.65 | 53.26 | 67.60 | 32.16 | 37.15 | 55.50 | 61.39 | 57.70 | 41.43 | 64.20 | 38.08 | 41.96 | 50.79 |
| IFD | 57.96 | 79.23 | 68.48 | 56.70 | 33.27 | 35.12 | 55.13 | 57.41 | 53.53 | 32.41 | 59.90 | 43.07 | 45.84 | 48.69 |
| SelectIT | 59.17 | 80.44 | 66.46 | 67.20 | 35.86 | 38.82 | 57.99 | 62.59 | 61.56 | 42.38 | 63.60 | 38.45 | 42.14 | 51.79 |
| Entropy | 61.30 | 55.04 | 61.04 | 68.90 | 37.34 | 40.48 | 54.02 | 58.61 | 50.72 | 44.02 | 61.40 | 32.90 | 37.89 | 47.59 |
| LESS$_{km}$ | 61.30 | 81.96 | 54.63 | 67.79 | 34.38 | 38.26 | **56.39** | 60.93 | 50.27 | 48.11 | 63.97 | 39.74 | 44.55 | **51.26** |
| IFD$_{km}$ | 60.19 | 78.77 | 59.70 | 66.81 | 30.31 | 31.79 | 54.60 | 60.74 | 58.98 | 40.37 | 62.95 | 40.67 | 42.70 | **51.07** |
| SelectIT$_{km}$ | 60.93 | 82.34 | 61.04 | 67.85 | 36.78 | 39.19 | **58.02** | 62.96 | 59.36 | 40.85 | 63.43 | 39.74 | 43.07 | 51.57 |
| Entropy$_{km}$ | 60.37 | 81.12 | 59.27 | 68.55 | 35.67 | 38.45 | **57.24** | 61.02 | 61.64 | 48.32 | 61.12 | 39.00 | 43.99 | **52.52** |

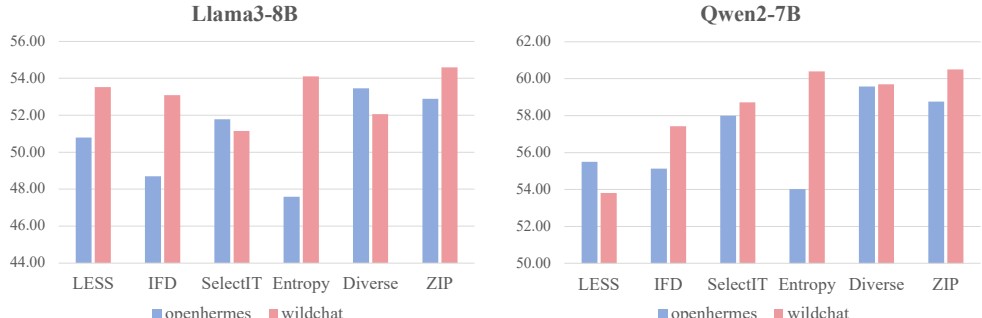

Figure 2: The average score (%) of each methods on Llama3 and Qwen2.

The lack of availability of **Less** is primarily evident in how its influence score is calculated. Since it requires computing the score for the final data point in the target task, it is essential to meticulously design a target set for each task to filter the data. However, in practical applications, we face a variety of training tasks that require our target data to be comprehensive and diverse. Hence, the effectiveness of LESS is strongly related to the quality of $\mathcal{D}_{val}$.

The **IFD** approach determines the ultimate IFD score by evaluating the perplexity (ppl) of the response. However, the length of the data significantly affects the ppl value. In particular, shorter data tend to produce excessively high ppl values, which contradicts with our expected results. Ultimately, we note that the IT data instructions selected by the IFD approach are quite brief, averaging merely 42 tokens on Openhermes, which aligns with the findings reported by Liu et al. (2023).

**SelectIT** can perform well at the model level, but it necessitates combining LLMs with various sizes to score the data. As IT datasets become larger, the computational cost required for LLMs with more parameters tends to increase exponentially, which limits their applicability to extensive datasets.

**Cross-entropy** is influenced by the length of responses. Typically, cross-entropy favors data with lengthy responses, whereas it shows no specific preference towards instructions. Consequently, the training samples will include simple instructions but extensive responses.

In addition, in this article, we do not use **NUGGETS** (Li et al., 2023d) as our baseline method. During our experimentation, we discover that the computational time for NUGGETS is significantly higher compared to other methods. Even with 40 A100 80G GPUs, it requires over 2,000 hours to perform the calculations. Given this high time cost, we decide to abandon this method.

The diversity-based approach usually outperforms the quality-based selection methods, however, one main issue with the diversity-based approach is its time and memory consumption.

To reproduce **DiverseEvol**, we utilized 8 A100 80G resources and consistently performed 3 iterations. However, each iteration requires 1-2 days, totaling 5-7 days to choose the final training subset. When dealing with large-scale data sets, the results often fall within the random range, though optimal results occur sporadically. This may be due to modifications in our implementation to address

memory constraints during replication (see Section 4.3), which may have slightly diminished the method's performance.

In contrast, **ZIP** does not need GPU resources, but the computing process is greedy. It incrementally adds 100 data at a time to the final training subset. For large data scales, it takes approximately 7 days to select 50,000 data. In addition, ZIP serves as a data selection method that operates independently of the model, meaning that the selected data cannot be adaptively tuned on the basis of the model. As illustrated in Tables 1 and 3, the data chosen by ZIP in OpenHermes perform poorly in both Llama3-8B and Qwen2-7B, whereas the data selected in WildChat exhibit the best performance across these models.

Moreover, we attempt to utilize **DQ** (Zhou et al., 2023a) as our baseline method. However, DQ uses a submodular strategy to choose a subset by optimizing submodular gains within the feature space. When dealing with millions of data points, it requires more than 1TB memory resources. Eventually, we decide to forgo this approach.

### 5.4 WHICH METHOD IS THE BEST?

By examining the average results of all methods, we notice that the majority of methods perform better with WildChat as the data source compared to OpenHermes, as illustrated in Figure 2, which is rather unexpected. Nonetheless, from a quality perspective, WildChat's conversation data tends to be noisy, particularly since the context of multiple conversation rounds is sometimes unrelated, while OpenHermes's data quality should be substantially higher than WildChat. However, the performance of the same data selection methods on these two types of data contradicts with our expectations. It is observed that the average token length for WildChat data is 1142, whereas for OpenHermes data, it is 354. Drawing inspiration from the work of Shen (2024), we devise a new experiment concentrating on data selection by token length. Initially, we obtain $N$ clusters through the K-Means process and subsequently select a certain amount of data based on the token length from each cluster proportional to its size. The results are presented in Table 4.

Based on Table 4, it is evident that using token length as the criterion for data selection generally yields optimal results. Specifically, for Llama3, regardless of whether the data source is Open-Hermes or WildChat, the results are superior to those achieved by other methods. In addition, the average score on WildChat (55.51) surpasses that obtained by fine-tuning with the entire dataset (54.58). Since random selection may not ensure the best fine-tuning results, we believe that **selecting data by token length can stably obtain a relatively high training benefit, reduce the uncertainty caused by randomness, and reduce costs.** This approach is particularly beneficial for BASE language models which generally have limited capabilities, as they tend to derive the most significant benefits from training on longer texts.

## 6 CONCLUSION

In this study, we observe that many SFT data selection methods depend on small-scale data sets, which do not meet the actual needs in real-world scenarios. This finding makes us rethink whether SFT data selection methods can work when they are required to handle large-scale IT datasets. We reproduce some existing self-scoring data selection approaches that do not need external LLMs' support on two million-scale datasets and find that almost all present methods do not significantly surpass random selection when dealing with large-scale datasets. Moreover, our analyses show that during the SFT phase, data diversity in data selection plays a more significant role than data quality. In addition, using token length as the quality metric is more appropriate for SFT data selection compared to other carefully crafted quality metrics.

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

# A APPENDIX

In this section, table 6, 7 includes training results of various methodologies with a training dataset comprising 50,000 entries 6, 7.

Table 6: The comprehensive results (%) on various downstream tasks using OpenHermes. Mention that CODE means Humaneval. Algorithm$_{km}$ means the algorithm has a Kmeans process, and Random$_x$ denotes the $_x$th random selection. The bold numbers indicate the best avg score of each part, and the underlined numbers indicate the second highest score.

| | Qwen2-7B | | | | | | | Llama3-8B | | | | | | |
| | BBH | GSM | CODE | MMLU | IFEVAL | | AVG | BBH | GSM | CODE | MMLU | IFEVAL | | AVG |
| | 3 shot | 8 shot | pass 1 | 5 shot | strict | loose | | 3 shot | 8 shot | pass 1 | 5 shot | strict | loose | |
| Base | 59.07 | 72.40 | 55.67 | 70.20 | 28.84 | 31.24 | 52.90 | 60.93 | 55.12 | 37.59 | 65.30 | 19.41 | 21.07 | 43.24 |
| all data | 61.39 | 80.12 | 63.32 | 68.50 | 40.85 | 44.18 | 59.73 | 63.33 | 73.24 | 46.43 | 63.90 | 46.40 | 49.72 | 57.17 |
| Random$_1$ | **62.87** | 80.67 | 62.44 | 68.33 | 34.75 | 38.08 | 57.86 | 63.89 | 64.37 | 46.19 | 62.75 | 45.10 | 49.72 | 55.34 |
| Random$_2$ | 61.11 | 80.82 | 65.76 | 68.12 | 38.08 | 40.67 | 59.09 | 62.13 | **66.57** | 47.32 | 61.57 | 46.58 | 49.54 | **55.62** |
| Random$_3$ | 61.02 | 81.35 | 60.15 | 68.54 | 38.63 | 40.85 | 58.42 | **65.65** | 63.53 | 44.05 | 61.96 | 42.51 | 46.21 | 53.99 |
| Random$_4$ | 60.37 | 80.06 | 55.98 | 68.95 | 37.34 | 40.30 | 57.17 | 62.78 | 62.40 | 45.12 | 62.41 | **47.87** | 50.83 | 55.24 |
| Random$_5$ | 60.19 | 80.14 | 63.29 | **69.16** | 38.08 | 40.85 | 58.62 | 64.72 | 65.13 | 45.18 | 62.51 | 45.47 | 49.17 | 55.36 |
| LESS | 60.46 | 80.29 | 58.66 | 67.40 | 39.00 | 43.25 | 58.18 | 61.02 | 57.85 | 17.01 | 63.01 | 40.30 | 46.40 | 47.60 |
| IFD | 57.50 | 80.52 | 67.13 | 66.79 | 35.86 | 38.08 | 57.65 | 61.94 | 52.84 | 44.63 | 63.36 | 41.04 | 43.99 | 51.30 |
| SelectIT | 60.56 | 79.98 | 62.77 | 67.96 | 36.04 | 39.00 | 57.72 | 61.20 | 64.22 | 40.03 | 62.40 | 41.96 | 44.92 | 52.46 |
| Entropy | 60.83 | 77.56 | 59.24 | 69.02 | 36.78 | 39.56 | 57.17 | 60.65 | 55.50 | **49.02** | 57.51 | 47.13 | **51.02** | 53.47 |
| Diverse | 61.67 | 81.35 | 61.89 | 68.60 | **44.55** | **46.40** | **60.74** | 63.33 | 61.11 | 48.75 | **63.62** | 46.21 | 49.17 | 55.37 |
| zip | 59.81 | 82.03 | **68.48** | 68.08 | 35.67 | 38.26 | 58.72 | 63.89 | 57.92 | 42.65 | 62.58 | 43.25 | 46.95 | 52.87 |
| LESS$_{km}$ | 61.20 | 81.88 | 54.51 | 67.77 | 32.90 | 36.60 | 55.81 | 61.02 | 59.44 | 47.04 | 63.35 | 42.14 | 47.32 | 53.39 |
| IFD$_{km}$ | 59.81 | 78.92 | 60.55 | 67.09 | 28.65 | 31.24 | 54.38 | 63.43 | 63.23 | 43.41 | 61.19 | 40.11 | 43.81 | 52.53 |
| SelectIT$_{km}$ | 61.20 | 81.20 | 66.52 | 69.10 | 34.57 | 38.45 | 58.51 | 61.85 | 61.49 | 45.76 | 61.64 | 43.44 | 48.43 | 53.77 |
| Entropy$_{km}$ | 61.02 | 80.82 | 66.04 | 68.25 | 36.78 | 39.37 | 58.71 | 61.85 | 64.22 | 48.66 | 61.85 | 42.70 | 46.58 | 54.31 |
| Length$_{km}$ | 60.46 | **83.62** | 63.35 | 68.79 | 38.26 | 41.59 | 59.35 | 65.09 | 62.70 | 47.29 | 62.73 | 45.10 | 49.17 | 55.35 |

Table 7: The comprehensive results (%) on various downstream tasks using WildChat. Mention that CODE means Humaneval. Algorithm$_{km}$ means the algorithm has a Kmeans process, and Random$_x$ denotes the $_x$th random selection. The bold numbers indicate the best avg score of each part, and the underlined numbers indicate the second highest score.

| | Qwen2-7B | | | | | | | Llama3-8B | | | | | | |
|---|---|---|---|---|---|---|---|---|---|---|---|---|---|---|
| | BBH | GSM | CODE | MMLU | IFEVAL | | AVG | BBH | GSM | CODE | MMLU | IFEVAL | | AVG |
| | 3 shot | 8 shot | pass 1 | 5 shot | strict | loose | | 3 shot | 8 shot | pass 1 | 5 shot | strict | loose | |
| Base | 59.07 | 72.40 | 55.67 | 70.20 | 28.84 | 31.24 | 52.90 | 60.93 | 55.12 | 37.59 | 65.30 | 19.41 | 21.07 | 43.24 |
| all data | 62.87 | 80.82 | 62.84 | 68.70 | 45.84 | 48.80 | 61.65 | 63.70 | 56.94 | 47.44 | 63.30 | 46.40 | 49.72 | 54.58 |
| Random$_1$ | 61.85 | 81.50 | 60.55 | 68.02 | 40.48 | 42.70 | 59.18 | 63.61 | 55.72 | 48.90 | 64.07 | 42.51 | 45.66 | 53.41 |
| Random$_2$ | 60.74 | 82.03 | 58.72 | 68.05 | 40.67 | 44.36 | 59.10 | 61.76 | 54.66 | 50.95 | 63.38 | 42.88 | 46.03 | 53.28 |
| Random$_3$ | 59.07 | 81.35 | 64.45 | 67.63 | 41.77 | 44.92 | 59.87 | 63.98 | 55.42 | **53.11** | 63.33 | 43.81 | 46.77 | 54.40 |
| Random$_4$ | **62.41** | 82.34 | 60.95 | 68.43 | 42.51 | 45.10 | 60.29 | 63.70 | 58.91 | 50.09 | 63.84 | 43.62 | 46.03 | 54.37 |
| Random$_5$ | 61.30 | 82.49 | 59.05 | 67.60 | 42.70 | 44.92 | 59.68 | 64.54 | 55.65 | 49.91 | 64.16 | 42.70 | 45.84 | 53.80 |
| LESS | 58.80 | 81.35 | 66.95 | 68.10 | 41.04 | 43.99 | 60.04 | 63.43 | 57.01 | 50.43 | **64.50** | 40.85 | 44.92 | 53.52 |
| IFD | 59.44 | 81.50 | 66.46 | 67.90 | 38.45 | 40.85 | 59.10 | 63.33 | 59.29 | 47.16 | 64.60 | 40.30 | 43.81 | 53.08 |
| SelectIT | 60.74 | **84.23** | 60.49 | **69.24** | 41.04 | 44.36 | 60.02 | 61.48 | 53.22 | 46.01 | 63.20 | 40.11 | 42.88 | 51.15 |
| Entropy | 61.02 | 81.96 | 60.88 | 68.40 | 43.07 | 46.58 | 60.32 | 61.48 | 55.34 | 48.90 | 64.02 | 47.50 | **51.02** | 54.71 |
| Diverse | 59.81 | 82.03 | 67.10 | 68.00 | 41.77 | 44.36 | 60.51 | **65.09** | 56.18 | 38.81 | 63.03 | 44.36 | 47.13 | 52.43 |
| zip | 59.91 | 79.83 | 71.04 | 67.97 | 42.88 | 45.84 | **61.25** | 64.72 | 57.16 | 41.49 | 61.54 | **45.84** | 48.43 | 53.20 |
| LESS$_{km}$ | 59.54 | 80.89 | 67.84 | 68.20 | **43.62** | **46.95** | 61.17 | 61.94 | 54.74 | 48.99 | 64.10 | 43.99 | 46.95 | 53.45 |
| IFD$_{km}$ | 59.26 | 80.67 | **68.41** | 68.13 | 41.77 | 43.99 | 60.37 | 62.69 | 56.10 | 48.63 | 63.02 | 40.85 | 42.70 | 52.33 |
| SelectIT$_{km}$ | 60.46 | 83.17 | 59.39 | 68.79 | 39.93 | 43.07 | 59.14 | 61.20 | 54.89 | 45.88 | 63.50 | 43.99 | 48.06 | 52.92 |
| Entropy$_{km}$ | 60.93 | 82.79 | 59.82 | 67.01 | 39.19 | 42.14 | 58.65 | 63.06 | 58.45 | 45.73 | 63.85 | 41.04 | 45.10 | 52.87 |
| Length$_{km}$ | 61.30 | 79.76 | 59.76 | 68.19 | 42.88 | 45.29 | 59.53 | 62.41 | **60.05** | 49.82 | 64.23 | 45.47 | 48.80 | **55.13** |

