# OpenReview forum: "Rethinking Data Selection at Scale: Random Selection is Almost All You Need"
_ICLR.cc/2025/Conference — Submitted to ICLR 2025_

### Official Review · Reviewer_MLme · 2024-10-29

**Soundness:** 2
**Presentation:** 2
**Contribution:** 2
**Rating:** 3
**Confidence:** 4

**Summary:**

This paper investigates supervised fine-tuning (SFT) for large language models (LLMs), specifically focusing on selecting a representative subset of training data that performs as well as or better than using the full dataset. It reveals that existing data selection methods, particularly self-scoring techniques, struggle to outperform random selection on million-scale datasets and highlights the importance of diversity over mere data quality in large-scale scenarios. Additionally, the authors find that filtering data by token length enhances SFT, especially for weaker base models like Llama3, offering a stable and efficient improvement method.

**Strengths:**

The finding that "Most self-scoring data selection techniques do not significantly outperform random selection on large-scale datasets" could contribute novel insights to the community.

**Weaknesses:**

- The motivation for this research is unclear. The authors claim that "This discrepancy creates a gap between the present SFT data selection strategies and real-world applications," yet it remains unclear what these real-world applications are.

- Finding 2, namely, "Data quality-based selection methods are more effective than data diversity-based methods when dealing with a small-scale dataset from a single source," is interesting, but exploring why this discrepancy occurs would be more valuable.

- Finding 3, that "token length could be used for data filtering," does not fit well within the overall narrative. In general, while the paper presents three findings, they are neither well-motivated nor well-connected. As an exploratory paper, its contribution feels limited.

- When using the term “significantly better,” please provide a p-value to substantiate the statistical significance of the results.

- Clarification:

  - The introduction should clarify what “scoring” means in this context and better motivate the need for scoring. Additionally, for the self-scoring approaches, the authors further categorise their techniques into two types: data quality-based methods and data diversity-based methods, but this distinction needs clearer explanation in the context.

    Although further detail is provided in Section 3, the introduction is challenging to follow.

  - In Figure 1, "BBH benchmark" requires a reference.

  - Line 58: "in practical applications" needs clarification.

- Typos:

  - Line 39: missing a space before the citation.
  - Table 1: some of the best numbers are not in bold.
  - Line 324: missing a space before the bracket.

**Questions:**

- What real-world applications could benefit from the outcomes of this research?
- What is the difference between data quality-based methods and data diversity-based approaches?
- Why is utilising token length in SFT beneficial for weaker base language models, like Llama3-8B?

---

> ### Author Response · Authors · 2024-11-22
> **(1/2)**
>
> Thanks for your helpful comments, below we will address your concerns point by point:
>
> **W1 & Q1:** We find that previous data selection methods were carried out on small-scale data sources, such as the alpaca data (with 5,200 pieces of data). However, the open-source models, like Llama3 or Qwen2, can not complete the SFT process with just tens of thousands of data instances. When training instruct models, a much larger scale of data (hundreds of thousands or even millions) is often required. For example, Qwen2 utilized over 500,000 pieces of data during the Supervised Fine-Tuning (SFT) process.[1] . Therefore, conducting data selection on only tens of thousands or a hundred thousand data sources is a rather toy scenario. In order to endow the model with the ability to effectively follow instructions and improve the reasoning abilities of LLMs, it is often necessary to train the model with large-scale data. Thanks to the reviewer's suggestions. We will make corresponding modifications in the subsequent versions.
>
> **W2:** When the data source is single, the homogenization of the data is rather serious. Especially when the scale of the dataset is small, in this case, there is actually no big difference between diversity-based selection and random selection. Therefore, selecting data by data quality can pick out higher-quality data.
>
> **W3:** Our findings were obtained step by step through our experimental analysis. Firstly, our motivation is that we want to explore the differences in effects between different methods and random selection, and we found that random selection is actually better when facing large-scale data sources (taking into account effectiveness and efficiency). Then, we arrived at our findings2 by comparing the effects between the quality-based method and the diversity-based method. Finally，at the beginning of section 5.4, when we compared the results of the openhermes data source and the wildchat data source, we found that although the data quality of openhermes was higher, the effect of wildchat was better instead. Therefore, we discovered that long texts have a more obvious benefit for the model. So we further proposed a method of selecting data by token length. Later, from the experimental results, compared with Qwen2-7B (a strong LLM), the gain was more obvious for llama3-8B (a weak LLM). Thus, we concluded that long text training is more suitable for weak LLMs. (In summary, our empirical findings are yield step-by-step, well-motivated, and well-connected). Since random selection may not ensure the best fine-tuning results, we believe that screening by token length can avoid the uncertainties brought by random selection and at the same time obtain a relatively good training result.
>
>
> [1] Yang A, Yang B, Hui B, et al. Qwen2 technical report[J]. arXiv preprint arXiv:2407.10671, 2024.

---

> ### Author Response · Authors · 2024-11-22
> **(2/2)**
>
> **W4:** We are grateful to the reviewers for their criticisms and corrections. We will revise the relevant modifiers to make them more rigorous. For the comparison of the effects between different baseline methods and the random method, we will add the corresponding significance test results in the subsequent versions.
>
> - We conducted the Mann - Whitney U test for each method against the results of 5 rounds of random selection. We adopted the right-tailed test approach, with the testing hypothesis being that the scores of each baseline method on different test tasks are greater than those of the random method. We also reported the p - value for each method being significantly better than that of the random method. We found that the p - value of all methods is higher than 0.05, which indicates that the results of all baseline methods are not greater than those of the random method.
>
> ||||||
> |-|-|-|-|-|
> |P-vlaue|llama3-openhermes|llama3-wildchat|Qwen2-openhermes|Qwen2-wildchat|
> |LESS|0.77|0.45|0.80|0.86|
> |IFD|0.85|0.53|0.85|0.68|
> |SelectIT|0.71|0.79|0.60|0.58|
> |Entropy|0.92|0.46|0.78|0.30|
> |Diverse|0.39|0.58|0.37|0.45|
> |zip|0.55|0.36|0.42|0.31|
>
> **Clarification&Typos**： Thanks to the reviewer's suggestions. We will make corresponding modifications in the subsequent versions.
>
> **Q2:** For the classification criteria based on quality and diversity, we referred to Paper [2] and provided an explanation in lines 132 - 135 and line 218-219. The quality-based approach means that the method focuses more on designing an algorithm and evaluation metrics to calculate the score of each piece of data, and finally conducts screening according to the data scores. The diversity-based method, on the other hand, focus more emphasis on the diversity of the dataset. Thank you for your suggestions. We will add this part of the explanation in the Intro or Related Work section in the subsequent versions.
>
> **Q3:** On the one hand, our experimental results indicate that long texts bring higher gains to Llama3. Moreover, the performance of the Llama3 series of models is somewhat inferior to that of the Qwen2 series[1]. On the other hand, in terms of the token length during the pre-training stage of the models, the pre-training token length of Llama3-8b is 8,192 [2], while that of Qwen2-7B is 32,768 [1]. Therefore, training Llama3 with long texts yields higher benefits. Besides, a longer token length means an increase in the overall training volume. Moreover, most of the response parts in the current SFT datasets are written by GPT-4. When the gap between the two models（GPT-4 and Llama3）is large, when we use the output of a more powerful model for training, the less capable the model is, the higher the gains it can obtain.
>
> [1] Yang A, Yang B, Hui B, et al. Qwen2 technical report[J]. arXiv preprint arXiv:2407.10671, 2024.
>
> [2] Qin Y, Yang Y, Guo P, et al. Unleashing the power of data tsunami: A comprehensive survey on data assessment and selection for instruction tuning of language models[J]. arXiv preprint arXiv:2408.02085, 2024.
>
> [3] Dubey A, Jauhri A, Pandey A, et al. The llama 3 herd of models[J]. arXiv preprint arXiv:2407.21783, 2024.
>
> **We have already submitted the revised paper and made corresponding modifications in response to the comments of all the reviewers.**

---

> > ### Comment · Reviewer_MLme · 2024-11-26
> > **Thank you for your detailed response**
> >
> > Thank you for your detailed response to the review comments and the additional clarifications. While I believe the paper can be further improved, some weaknesses persist. For instance, the motivation for the task remains unclear, the results do not fully support the conclusions, and the conclusions lack novelty compared to existing work. I have no further questions and will maintain my original rating.

---

### Official Review · Reviewer_mJQN · 2024-10-31

**Soundness:** 3
**Presentation:** 3
**Contribution:** 3
**Rating:** 5
**Confidence:** 4

**Summary:**

In this paper, the authors study the effectiveness of data selection methods on large-scale datasets, which is largely unexplored. With experiments on large-scale datasets, the authors find that nearly existing data selection methods do not significantly outperform random selection on large-scale datasets. This paper also tries to explain the reason that leads to inferior performance of existing methods on a larger data pool. Ultimately, the authors propose a more effective data selection method based on token length. This paper offers researchers a new perspective on data selection in large language models.

**Strengths:**

The authors study the effectiveness of data selection methods on large-scale datasets, and find the inferior performance of existing methods when dealing with large-scale data pools. At the same time,  the authors try to analyze the reason that leads to inferior performance of existing methods on a larger data pool.

**Weaknesses:**

1. **Title Reflecting Content**: The title of this paper suggests that ”random
selection is almost all you need”, while results show that the diversity-based
selection method achieves superior performance with an average score of
59.58 under Qwen2-7B in Table 1. The result does not indicate that random
selection is a better method. In this paper, it seems to show that
diversity plays an important role in data selection for large-scale datasets. A more representative title improves clarity
and helps set readers’ expectations.

2. **Insufficient experiments**: This paper’s results show that existing data
selection methods do not significantly outperform random selection on large-scale datasets. However, the authors only perform experiments on two models. Are there similar outcomes for different LLMs (e.g., different-sized models)?

**Questions:**

**Questions**: This paper proposes a data selection method by token
length to achieve optimal results. However, as shown in Table 3, does the Qwen2-7B model not get better results with this method compared with
existing methods? By the way, why is this selection method beneficial? In
other words, what is the motivation for this approach? Moreover, what
does it mean to ”reduce the uncertainty caused by randomness”?

---

> ### Author Response · Authors · 2024-11-22
>
> Thanks for your helpful comments, below we will address your concerns point by point:
>
> **W1:** We acknowledge that the effectiveness of random selection on Qwen2 - 7B is not the best. However, we hold the view that the criterion for determining the applicability of a method does not necessarily depend on the model's training outcomes. Although the final outcome of DiverseEvol is approximately 1.5% higher than the average effect of random selection (58.11), it takes 5 to 7 days for DiverseEvol to finish the entire data selection process. Here, a tradeoff exists between the effectiveness and time efficiency. The significant increase in time cost for a relatively small improvement in effectiveness makes such a time investment in DiverseEvol perhaps not worthwhile. Therefore, after comprehensively considering the balance between effectiveness and efficiency, we concluded that when dealing with large - scale data sources, random selection is the most advantageous option.
>
> Besides, We conducted the Mann - Whitney U test for each method against the results of 5 rounds of random selection. We adopted the right-tailed test approach, with the testing hypothesis being that the scores of each baseline method on different test tasks are greater than those of the random method. We also reported the p - value for each method being significantly better than that of the random method. We found that the p - value of all methods is higher than 0.05, which indicates that the results of all baseline methods are not greater than those of the random method.
>
> ||||||
> |-|-|-|-|-|
> |P-vlaue|llama3-openhermes|llama3-wildchat|Qwen2-openhermes|Qwen2-wildchat|
> |LESS|0.77|0.45|0.80|0.86|
> |IFD|0.85|0.53|0.85|0.68|
> |SelectIT|0.71|0.79|0.60|0.58|
> |Entropy|0.92|0.46|0.78|0.30|
> |Diverse|0.39|0.58|0.37|0.45|
> |zip|0.55|0.36|0.42|0.31|
>
> **W2:** Most of the previous work has been tested on LLMs with a scale of 7B or 14B. Considering the amount of our data and the time cost of the baseline methods, we finally only conducted experiments on two 7B models. In fact, during the data selection stage, we carried out data processing work simultaneously with 8 A100 GPUs. If we want to select 50K samples, IFD takes 1.5 days to complete the data processing of one model on one dataset, LESS needs 1 day, Select requires 1 day, cross entropy takes 0.5 days, ZIP needs about 7 days, and DiverseEvol on average takes 6 days. The whole process takes about 17 days (on the premise that there is no task queuing waiting involved). In the case of two models and two datasets, we need a longer time to reproduce all the baseline methods. Therefore, we did not conduct experiments on the scale of 14B in the main text.
>
> Considering the time cost, we added some experiments on selectIT and ZIP based on Qwen2.5-3B and Qwen2.5-14B, as shown in the following table. Judging from the results, the performance of ZIP is about 0.8% higher than that of random（avg）, and is not significantly higher than random.
>
> ||bbh|gsm8k|humaneval|mmlu|IFEVAL||avg|
> |-|-|-|-|-|-|-|-|
> ||3 shot|8 shot|pass 1|5 shot|strict|loose||
> |Qwen2.5-14B-select|72.04|88.86|78.72|78.86|49.72|54.53|70.46|
> |Qwen2.5-14B-zip|72.96|87.79|77.71|77.62|53.79|57.12|71.17|
> |Qwen2.5-14B-openhermes-random-2|73.8|83.55|74.27|77.57|54.16|57.12|70.08|
> |Qwen2.5-14B-openhermes-random-3|74.17|88.7|74.21|77.87|52.31|55.64|70.48|
> |Qwen2.5-14B-openhermes-random-5|73.52|87.79|75.7|77.03|52.5|56.93|70.58|
> |random-avg|73.83|86.68|74.73|77.49|52.99|56.56|70.38|
>
> ||bbh|gsm8k|humaneval|mmlu|IFEVAL||avg|
> |-|-|-|-|-|-|-|-|
> ||3 shot|8 shot|pass 1|5 shot|strict|loose||
> |Qwen2.5-3B-select|55.37|77.56|56.86|64.89|33.46|36.41|54.09|
> |Qwen2.5-3B-zip|56.11|77.63|56.83|64.54|36.23|37.89|54.87|
> |Qwen2.5-3B-openhermes-random-1|55.56|78.39|57.1|64.19|34.94|36.23|54.40|
> |Qwen2.5-3B-openhermes-random-2|53.24|75.97|56.16|64.71|34.2|36.23|53.42|
> |Qwen2.5-3B-openhermes-random-3|55.83|77.1|59.24|64.81|30.31|32.9|53.37|
> |random-avg|55.22|77.33|57.24|64.63|33.83|35.93|54.03|
>
> **Q1:** At the beginning of section 5.4, when we compared the results of the openhermes data source and the wildchat data source, we found that although the data quality of openhermes was higher, the effect of wildchat was better instead. Therefore, we discovered that long texts have a more obvious benefit for the LLM SFT. So we further proposed a method of selecting data by token length. Later, from the experimental results, compared with Qwen2-7B (a strong LLM), the gain was more obvious for llama3-8B (a weak LLM). Thus, (we conclude that) long text training is more suitable for weak LLMs. Since random selection may not ensure the best fine-tuning results, we believe that screening by token length can avoid the uncertainties brought by random selection and at the same time obtain a relatively good training result.
>
> **We have already submitted the revised paper and made corresponding modifications in response to the comments of all the reviewers.**

---

> > ### Comment · Reviewer_mJQN · 2024-11-26
> >
> > Thank you for the detailed response. I conjecture the main idea that the authors want to express is that existing data-selecting methods can not significantly outperform random selection when dealing with large-scale datasets. If the authors consider the tradeoff of methods, highlighting the point of the tradeoff in this paper will help readers understand this work better. In addition, the authors find a more effective data selection method based on token length. I am interested in the performance of this method on more general datasets (**Don't conduct experiments** ). Could you share your thoughts on whether your time is okay?

---

> > > ### Author Response · Authors · 2024-11-26
> > >
> > > Thank you for your comment and suggestions. We will highlight our concern about the trade-off between effectiveness and efficiency in the new version.
> > >
> > > Additionally, regarding whether the method of selecting data by length you mentioned is applicable to other general datasets, we believe that selecting data by length is also effective on most general datasets. On the one hand, in Direct Preference Optimization (DPO) paper [1], researchers have found that LLMs are more inclined to output long texts. Moreover, the previous work DEITA [2] also holds that complex, difficult, and long data samples are more beneficial for  SFT. On the other hand, we have discovered that both the pre-training parts of Qwen 2 [3] and Llama 3 [4] include long-text training sections. Therefore, we think that long-text training is a means to improve the performance of LLMs. Consequently, select data by length should be able to improve the model's performance to some extent on most datasets.
> > >
> > > [1] Rafailov, R., Sharma, A., Mitchell, E., Ermon, S., Manning,C. D., and Finn, C. Direct preference optimization: Your language model is secretly a reward model. arXiv preprint arXiv:2305.18290, 2023.
> > >
> > > [2] Wei Liu, Weihao Zeng, Keqing He, Yong Jiang, and Junxian He. What makes good data for alignment? a comprehensive study of automatic data selection in instruction tuning. arXiv preprint arXiv:2312.15685, 2023.
> > >
> > > [3] Yang A, Yang B, Hui B, et al. Qwen2 technical report[J]. arXiv preprint arXiv:2407.10671, 2024.
> > >
> > > [4] Dubey A, Jauhri A, Pandey A, et al. The llama 3 herd of models[J]. arXiv preprint arXiv:2407.21783, 2024.

---

> > > > ### Comment · Reviewer_mJQN · 2024-11-27
> > > >
> > > > Thank you for the detailed response and answer to the questions.  I acknowledge that the existing data-selection methods exhibit poor performance on large-scale datasets. However, this paper still lacks deeper novelty and a clear structure. Moreover, the experimental results can not strongly support all conclusions. Thus, I chose to maintain my origin rating.

---

### Official Review · Reviewer_HChF · 2024-11-03

**Soundness:** 2
**Presentation:** 2
**Contribution:** 3
**Rating:** 5
**Confidence:** 4

**Summary:**

This paper studies the efficacy of various data filtering strategies when applied to large-scale SFT datasets (i.e.., OpenHermes2.5, WildChat). While these same strategies often outperform random selection on smaller-scale SFT datasets they are shown to be surprisingly  less impactful when applied to these larger-scale (state-of-the-art) open-source datasets. On these datasets, approaches that target diversity seem to be more effective than those that target quality. Finally, the authors test their own length-based filtering strategy, which outperforms the other methods when the base model is Llama3-8B.

**Strengths:**

**Originality and Significance.** This paper performs a valuable and novel (to the best of my knowledge) empirical study to see whether existing SFT data selection methods remain effective when applied on larger SFT datasets. Because these larger SFT datasets are generally better than the smaller datasets these methods were previously tested on, it is important to see whether filtering works in this more state of the art regime. The main findings that (i) many of these methods no longer are as impactful; (ii) diversity is more important than quality-filtering are therefore quite significant and potentially highly-relevant to future SFT data curation efforts.

**Quality.** This paper implements a good number of baselines and also is thorough about trying multiple base models and dataset sizes (post data selection). These variations are nice to test the robustness of the main takeaways.

**Clarity.** Overall, most of the paper is easy to follow with the exception of the description of baseline methods (see W3).

**Weaknesses:**

**(W1) Isolating dataset size.** A central claim in the paper is that dataset size is what causes differences in the behavior of the baselines. However, there are also other differences between OH2.5/WildChat and older datasets besides data quantity; e.g., quality-based selection may be less impactful simply because these larger datasets are already the result of more careful curation and thus higher-quality compared to older ones. An experiment that would better isolate data quantity (and control for quality) would be to run the existing experiments on progressively smaller random subsets of OH2.5/WildChat (e.g., what happens when the starting pool is 10K/100K examples of OH2.5 v.s. 1M?)

**(W2) Understanding run-to-run variance / Fairness of comparison to random.** The stated takeaway from the results is that *"it is more efficient to randomly select training data...random selection reduces costs and yields superior training results"* but I don't know if that's fully borne out by the results.
- For the Qwen2 results (which achieve highest overall perforamnce), in the 50K regime, the best average results are achieved by non-random curation methods (Diverse and ZIP each score 1% higher than the max over the random runs in Tables 5 and 6 respectively). In the 10K regime, this is also almost true with the exception of 1/5 of the random runs.
- In general, I'm not sure its fair to compare the max over 5 random runs to a single run of a selection method. if only 1 of 5 runs ends up doing better, then in practice you'd likely have to try multiple times when randomly sampling a subset that outperforms non-random selection, increasing costs.
- Ideally, it would be nice to show the average and stdev across the 5 random runs and to also get a sense of run-to-run variation for the non-random methods (though I acknowledge this might be costly).

**(W3) Clarity when describing existing methods.** I currently found Sec 3. a bit hard to follow. It may be helpful to define up front unified notation for variables that appear in multiple methods instead of using the differing notation from each of the original papers (e.g. whether a sample is given by $z$ or $x$ or $n$). Also, in some places, symbols are currently used without an explicit definition (e.g. $\hat{\Gamma}$ in LESS, $\mathcal{C}$ in ZIP).

**Miscellaneous Notes**
- In captions for Tables 1 and 2, "the suboptimal score" is confusing terminology, I believe you meant "second highest score"?
- Given that originally LESS is a targeted selection method, I'm not sure it's completely aligned with the spirit of the original method to set the target as a subset of the overall data pool? I suspect what might be leading to bad performance is that this may actually result in selecting a less diverse dataset (by prioritizing examples that represent the majority).

**Questions:**

(Q1) Could you share more details about Figure 1, given that the difference between small/large regimes is a key aspect of the paper? In particular, which datasets were used for the 10K-300K scale in Figure 1? Are these already random subsets of OH2.5/WildChat (basically what i suggested in W1) or are they some other datasets (e.g. Dolly, ShareGPT)?

(Q2) What is the cost of the fine-tuning runs in your experiment setup? It would be helpful to know this relative to the costs of running the various data curation algorithms (when discussing them in 5.3).

---

> ### Author Response · Authors · 2024-11-22
> **(1/2)**
>
> Thanks for your helpful comments, below we will address your concerns point by point:
>
> **W1:**  We randomly selected 100k pieces of data on OpenHermes. Based on this, we further selected 10k pieces from this 100k data using different baseline methods as the training set for SFT. The results are shown in the following table. Compared with selecting from the full set, when selecting from 100k pieces of data, the quality-based method can achieve a certain improvement in performance. However, the performance of the diversity-based method has slightly declined. We believe this is because changing our data source from 1M to 100k has affected the data distribution of the data source to some extent, resulting in a slight decline in the effectiveness of the diversity-based method on this subset. We found that when the scale of the dataset becomes smaller, some of our baseline methods are only slightly better than the results of random selection. This might be due to the fact that the dataset we used is more diverse than the datasets used by these methods in their respective papers.
>
> |base on 10w data（OpenHermes）|bbh|gsm8k|humaneval|mmlu|IFEVAL||avg|
> |-|-|-|-|-|-|-|-|
> ||3 shot|8 shot|pass 1|5 shot|strict|loose||
> |random-avg|63.02|57.24|43.18|62.47|44.12|48.06|53.02|
> |llama3-8B-less|64.91|57.09|45|63.61|40.67|43.25|52.42|
> |llama3-8B-ifd|64.35|59.06|44.91|63.54|42.88|46.21|53.49|
> |llama3-8B-select|62.04|62.55|45.46|63.42|42.7|45.47|53.61|
> |llama3-8B-entropy|61.85|57.47|42.47|63.88|42.33|47.32|52.55|
> |llama3-8B-diverse|64.01|59.28|48.72|63|41.04|44.55|53.43|
> |llama3-8B-zip|63.33|59.51|41.83|61.75|39.93|43.44|51.63|
>
> |base on 100w data（OpenHermes）|bbh|gsm8k|humaneval|mmlu|IFEVAL||avg|
> |-|-|-|-|-|-|-|-|
> ||3 shot|8 shot|pass 1|5 shot|strict|loose||
> |random-avg|63.24|58.48|44.69|63.33|44.05|47.38|53.53|
> |llama3-8B-less|61.39|57.7|41.43|64.2|38.08|41.96|50.79|
> |llama3-8B-ifd|57.41|53.53|32.41|59.9|43.07|45.84|48.69|
> |llama3-8B-select|62.59|61.56|42.38|63.6|38.45|42.14|51.79|
> |llama3-8B-entropy|58.61|50.72|44.02|61.4|32.9|37.89|47.59|
> |llama3-8B-diverse|65|56.25|44.51|63.84|43.99|47.13|53.45|
> |llama3-8B-zip|63.98|59.67|40.7|62.6|43.81|46.58|52.89|
>
> **W2:**
>
> - Even if some methods perform well on a certain model, as you mentioned (Diverse and ZIP each score 1% higher than the max over the random runs in Tables 5 and 6 respectively), the time cost they require is still extremely high compared to random selection. When selecting 50k pieces of data, both of these methods take about seven days (using 8 A100 GPUs). Therefore, we believe that the random method is more suitable for data screening of large-scale data sources.
>
> - We presented the results of five rounds of random selection in the hope of demonstrating that the effectiveness of most of the current baseline methods is actually within the range of random selection. Combined with the results of the significance tests we will mention next, none of the current baseline methods are significantly better than random. Therefore, we believe that it is not necessary to conduct multiple random selections in practice.
>
> - We conducted the Mann - Whitney U test for each method against the results of 5 rounds of random selection. We adopted the right-tailed test approach, with the testing hypothesis being that the scores of each baseline method on different test tasks are greater than those of the random method. We also reported the p - value for each method being significantly better than that of the random method. We found that the p - value of all methods is higher than 0.05, which indicates that the results of all baseline methods are not greater than those of the random method.
>
> ||||||
> |-|-|-|-|-|
> |P-vlaue|llama3-openhermes|llama3-wildchat|Qwen2-openhermes|Qwen2-wildchat|
> |LESS|0.77|0.45|0.80|0.86|
> |IFD|0.85|0.53|0.85|0.68|
> |SelectIT|0.71|0.79|0.60|0.58|
> |Entropy|0.92|0.46|0.78|0.30|
> |Diverse|0.39|0.58|0.37|0.45|
> |zip|0.55|0.36|0.42|0.31|

---

> ### Author Response · Authors · 2024-11-22
> **(2/2)**
>
> **W3:** In LESS, $\tilde{\Gamma}(\boldsymbol{z},\theta)$ represents the gradient values of different data $\boldsymbol{z}$ under different optimization states $\theta$. In ZIP, $\mathcal{C}$ means the compression ratio. We are sorry to make you confused, and we will make adjustments in the subsequent versions according to your suggestions. Thank you for your suggestions.
>
> **Miscellaneous Notes 1:** yes， it means "second highest score"， we will make adjustments in the subsequent version.
>
> **Miscellaneous Notes 2:** In the original paper, LESS involves selecting a part of the data for each task to train an LLM specifically targeted at a particular task. In this paper, our aim is to select a subset that can be adapted to any tasks. Therefore, we have modified a part of the code. On the one hand, we believe it does not conform to the application scenarios of LLMs at the current stage, the goal of Supervised Fine-Tuning (SFT) should be to enable the model to unlock different capabilities to adapt to different downstream tasks. The original intention of LLM SFT is to improve the generalization ability of the model. Previous works [1, 2] involved selecting a subset of data from the source dataset and then conducting evaluations on different tasks. On the other hand, considering the time cost and resource consumption of data selection, we did not train a separate model for each task.
>
> **Q1:** In Figure 1, "Data size 10K - 300K" refers to the data sources used in the original texts of different methods. Among them, LESS uses the collection of FLAN V2 COT, DOLLY, and OPEN ASSISTANT 1 datasets, with a total of over 270,000 pieces of data. IFD and diverseEVol use the alpaca dataset with 5.2k data. SelectIT used alpaca-gpt4 datasets with 5.2k data. ZIP used 300K samples obtained from WizardLM, ShareGPT and UltraChat. "Data size 1M" refers to Openhermes2.5-1M dataset.
>
> **Q2:** During the fine-tuning stage, we conduct full-parameter fine-tuning using 8 A100 GPUs. If the number of fine-tuning data is 10,000 pieces, it will take approximately 30 minutes. If the number of fine-tuning data is 50,000 pieces, it will take about 2.5 hours. In fact, the time cost in the fine-tuning stage is actually quite low. The most significant one is the time cost in the data selection stage. During the data selection stage, if we want to select 50K samples, apart from the random method, the fastest method is cross entropy, which takes approximately 15 hours to process one million pieces of data simultaneously using 8 A100 GPUs. IFD takes 1.5 days to complete the data processing of one model on one dataset, LESS needs about 1 day, SelectIT requires about 20 hours, ZIP needs about 7 days, and DiverseEvol on average takes 6 days.
>
> [1] Wei Liu, Weihao Zeng, Keqing He, Yong Jiang, and Junxian He. What makes good data for alignment? a comprehensive study of automatic data selection in instruction tuning. arXiv preprint arXiv:2312.15685, 2023
>
> [2] Ming Li, Yong Zhang, Zhitao Li, Jiuhai Chen, Lichang Chen, Ning Cheng, Jianzong Wang, Tianyi Zhou, and Jing Xiao. From quantity to quality: Boosting llm performance with self-guided data selection for instruction tuning. arXiv preprint arXiv:2308.12032, 2023b
>
> **We have already submitted the revised paper and made corresponding modifications in response to the comments of all the reviewers.**

---

> > ### Comment · Reviewer_HChF · 2024-11-26
> > **Response and Additional Questions in response to Rebuttal**
> >
> > Thanks to the authors for the detailed response and additional results! I really appreciate these additions but do still have some concerns and questions.
> >
> > **Re: W1**, my biggest concern is with the main narrative of the current paper, which focuses on the difference in scale between older and newer datasets to explain the difference in behavior of selection methods. However the larger datasets you test on (e.g. OH2.5) differ not only in scale but also quality/diversity compared to older ones (e.g. Alpaca), likely already the result of some careful (human-driven) curation. I believe discussing these factors more is key to understanding the discrepancies in performance.
> >
> > * I appreciate the new results here where the scale of the OH starting pool is reduced to be similar to the datasets used in other works (e.g., from Fig. 1). But if I am reading the results correctly, the selection methods still do not help as much as they do when applied on top of older lower-quality datasets. In the "base on 100w data" table, none of the quality-based methods improve BBH whereas in Fig. 1, many of them do, even when the initial pool is larger (270K in the case of LESS). To me, this already seems indicative that size is not the only important factor.
> >
> > * Conversely, your conclusions about scale would be bolstered if you were able to show that selection methods stop working on a large but lower-quality dataset. A priori though, I'm skeptical that this would hold.
> >
> > * Perhaps more broadly, the important practical question is how to build better SoTA datasets compared ones like OH2.5/Wildchat (and ideally with as little human-effort as possible). I believe the current results do make a key contribution here, which is to show that **additional filtering of the these current SoTA datasets is unlikely to be helpful.** However, my feeling is that non-random selection could still be useful for filtering less-curated sources that are potentially larger and more diverse but have lower average quality (e.g., approaches that synthesize Q-A pairs from web-crawled data like Humpback). Indeed this could be one pathway for building a new dataset that does not start from OH2.5/Wildchat or could be used to augment them to further increase their diversity.
> >
> > **Re: W2**, the additional context about compute costs and p-values is indeed helpful to know (though I'm not sure if this then weakens conclusions about diversity-based methods being more effective than quality-based). Interestingly, it seems based upon **Q2**, most selection methods actually take at least as long as simply training on all 1M examples, which would be more effective for improving performance in most cases. To me, this seems like an important failure of these selection strategies and should be emphasized!
> >
> > **Re: Q1**, so to clarify, the blue bars in Fig. 1 correspond to different datasets depending on the method on x-axis and all the results were re-run with your implementations? IMO it might be cleaner to just standardize the initial pool across all methods or to at least mention explicitly what dataset corresponds to what bar. Also, to follow-up
> > * Any reason you measure BBH here instead of an average over all tasks
> > * Did you mean 52K instead of 5.2K for Alpaca?

---

### Official Review · Reviewer_yhuK · 2024-11-04

**Soundness:** 2
**Presentation:** 2
**Contribution:** 3
**Rating:** 3
**Confidence:** 4

**Summary:**

This paper investigates the effectiveness of data selection methods for supervised fine-tuning (SFT) of Large Language Models (LLMs) at scale. Through extensive experimentation on million-scale datasets, the research reveals that most existing self-scoring data selection methods, which don't require external model assistance, fail to significantly outperform random selection when applied to large-scale data pools.

**Strengths:**

1. This paper studies an interesting yet important task of selecting training data for supervised fine-tuning (SFT) in large language models.

2. The experimental results reveal that various data selection methods struggle to significantly outperform random selection, a finding that will likely be beneficial to practitioners.

**Weaknesses:**

1. Some of the conclusions presented are already established in existing literature. For instance, paper [1] has previously demonstrated the importance of data diversity in SFT data selection with large-scale datasets, while paper [2] has shown the effectiveness of selecting longer answers for data selection. It appears the authors overlooked these references, which limits the novelty and contribution of this work.

2. Although the paper identifies that certain data selection methods fail to outperform random selection, it lacks a thorough analysis—either theoretical or empirical—explaining why these approaches underperform. This absence of deeper exploration reduces the overall depth and insight provided by the study, which is required by top-tier ML conferences.

3. The paper does not provide a clear definition of what is an "extensive SFT dataset." In practical applications, it remains unclear how to determine whether a dataset qualifies as "extensive," which could lead to ambiguity in interpreting the paper's findings and recommendations.


[1] Bukharin, Alexander, and Tuo Zhao. "Data diversity matters for robust instruction tuning." arXiv preprint arXiv:2311.14736 (2023).

[2] Zhao, Hao, et al. "Long Is More for Alignment: A Simple but Tough-to-Beat Baseline for Instruction Fine-Tuning." Forty-first International Conference on Machine Learning.

**Questions:**

See above.

---

> ### Author Response · Authors · 2024-11-22
>
> Thanks for your helpful comments, below we will address your concerns point by point:
>
> **W1:** Our paper focuses more on the comparison between different methods. The main conclusion finally drawn is that when faced with large-scale data sources, the effectiveness of most data selection methods does not actually significantly exceed the results of random selection. The  first reference paper you mentioned proposed the QDIT method, but it only compared with random selection and did not compare with any state-of-the-art (SOTA) methods (and the first reference only talks about data diversity-based method, while our paper include discussions on both data diversity-based and data quality-based methods). Similarly, the second reference paper also did not compare with any SOTA methods and only conducted experiments on small-scale datasets.   In summary, our paper focus more on the experimental analysis of different methods based on large-scale data source, and thus draws conclusions. Although the conclusions are similar, our focus is different from that of these two articles.
>
> **W2:** In Section 5.3, we analyzed the unavailability of different methods from multiple aspects. Some aspects involve the inherent flaws of the methods themselves. For example, as pointed out in lines 414 to 415 of our article, the baseline method IFD tends to preferentially select data with shorter queries, which is consistent with the research findings reported in Paper [1]. In addition, some baseline methods are not suitable for data screening of large-scale data sources. For instance, as described in lines 429 to 430, the DiverseEvol method has a relatively high time cost (It needed average 6 days to finish the selection stage). Our experimental analysis comprehensively validates our claim that the random selection method is more suitable for selecting data from large-scale data sources.
>
> **W3:** We consider that large-scale data sources refer to datasets with a data volume at the million level and diverse data sources (For example, annotated by dedicated workers, sourced from real users, and synthesized with models) as well as rich data types (For example, code data, math data, conversations, knowledge Q&A, etc.). Thanks to the reviewers' suggestions, we will include this definition in the subsequent versions.
>
> [1] Wei Liu, Weihao Zeng, Keqing He, Yong Jiang, and Junxian He. What makes good data for alignment? a comprehensive study of automatic data selection in instruction tuning. arXiv preprint arXiv:2312.15685, 2023
>
> **We have already submitted the revised paper and made corresponding modifications in response to the comments of all the reviewers.**

---

> ### Comment · Reviewer_yhuK · 2024-11-24
>
> Thank you for your response and the additional clarification—it has helped me gain a better understanding of the paper. However, I still feel that the contribution of this work is somewhat limited. The analyses presented appear to be similar to those in existing works, offering limited novel insights or deep analysis. Additionally, there is no formal exploration of the relationship between the optimal strategy and the amount of data, which I believe is an important aspect to address. I choose to keep my score at this time, but encourage the authors to further improve the current manuscript.

---

> > ### Author Response · Authors · 2024-11-24
> >
> > We thank the reviewer for your comment.
> >
> > We still would like to emphasize again that our core viewpoint is that "when facing large-scale data sources, most baseline methods are not significantly better than random selection". In the paper, we focus more on the comparison among different methods. All of our conclusions are also drawn from the comparisons between models, which is different from the two papers you mentioned.

---

> ### Author Response · Authors · 2024-11-24
>
> Thank you for your comment. We'd like to add some more points.
>
> We believe that when judging the novelty of an article, it is not appropriate to take it out of context and it is not fair to assess the novelty of an article solely based on its partial contributions, but rather from the perspective of the full contributions.
>
> Regarding the evaluation tasks, the first paper and our paper mainly focus on eliciting LLMs' reasoning and code generation abilities via SFT, while the second article mainly focuses on the alignment task. Therefore, it is not reasonable to predict that the length-based data selection method can yield better results in SFT downstream tasks as there exists a length-based method that performs well in alignment tasks. In general, the differences between reasoning and alignment tasks are significant.
>
> From the perspective of motivation, the first article mainly seeks better data selection methods from the perspective of effectiveness, and the second article is inspired by the fact that GPT-4 prefers giving text with longer lengths higher scores in alignment evaluation. However, our paper is looking for data selection methods with better comprehensive performance from the perspective of the balance between effectiveness and efficiency.
>
> In terms of the analysis part, the first article only compared the data-diversity part and data-quality part of its own method on large datasets, but we compared several different diversity-based methods and quality-based methods on million-level SFT datasets, our analyses are more comprehensive. The second article only verified the performance of the proposed length-based data selection method on small-scale datasets by data slicing, but our proposed method was inspired through analyzing the characteristics of the WildChat and OpenHermes datasets in combination with the subsequent experimental analysis of the token length part. We simultaneously compared and analyzed the performance advantages and disadvantages of the data quality-based, data diversity-based, and random selection methods on large-scale datasets. We believe that in the era of large models, it is more meaningful to conduct experimental analysis on large-scale datasets.
>
> From the perspective of conclusion, the main contributions of these three papers are different. We obtained our three findings step by step through experimental analysis, and the way we conduct experiments is different from those used in the two references.
>
> The paper [1]‘s conclusion is: There is a natural tradeoff between data diversity and quality.
>
> The paper [2]'s conclusion is: A lightweight refinement of long instructions can further improve the alignment abilities of the LLMs.
>
> Our conclusions are:
> 1. Most self-scoring data selection techniques do not significantly outperform random selection on large-scale datasets.
> 2. Data diversity holds more significance than data quality during the SFT phase.
> 3. It is useful to utilize token length as a criterion to conduct data filtering, yielding stable and efficient results for SFT when dealing with large-scale IT data.
>
> In summary, our conclusions are more comprehensive.
>
> [1] Bukharin, Alexander, and Tuo Zhao. "Data diversity matters for robust instruction tuning." arXiv preprint arXiv:2311.14736 (2023).
>
> [2] Zhao, Hao, et al. "Long Is More for Alignment: A Simple but Tough-to-Beat Baseline for Instruction Fine-Tuning." Forty-first International Conference on Machine Learning.

---

### Official Review · Reviewer_2efs · 2024-11-04

**Soundness:** 3
**Presentation:** 3
**Contribution:** 2
**Rating:** 6
**Confidence:** 3

**Summary:**

In this paper authors proposed a data filtration technique for the SFT (supervised fine-tuning phase) of LLM training.
Rigorously testing existing SOTA methods that were proven to work on the small scale datasets, authors found that
filtering data by token length offers an efficient way for improving downstream results further. In addition to that authors demonstrated that SOTA methods fall short in quality when handling large-scale 1T datasets. Authors demonstrated that data diversity plays the most important role in the SFT phase, thus quality filtering is suboptimal.

**Strengths:**

Constructing a high-quality multi-corpora dataset that is useful for training LLMs is a hard and important task. Input data is crucial for making a high-quality reliable and safe LLM at scale. Authors explore the topic of improving the dataset construction process and suggest a simplified framework that does not require rigorous data cleaning at scale. Considering that developing and using filtration techniques at scale might present computational challenges, this paper makes a significant contribution to the field with their simplified proposal.

**Weaknesses:**

One of the challenges with assessing the quality of LLMs is the quality and generalizability of the downstream tasks. Majority of the times that amount of the overlap between training and downstream data can cause skewed assessments in the quality of data processing techniques. I could not find any statistics reported by the authors for the amount of the overlap present in their training corpuses and the downstream tasks. It is important to understand that before drawing any conclusions with regards to the data filtration techniques.
It might be the case that some of the filtration strategies are heavily biased towards the examples present in the downstream tasks, and not necessarily improve model's generazability qualities.

**Questions:**

I would like to see the stats on the overlap present in the tokens from your SFT datasets and downstream tasks. It would also be useful to know whether any deduplication method was applied on top of the SFT datasets to eliminate the effect of model's memorization.

---

> ### Author Response · Authors · 2024-11-22
>
> We are very grateful for the suggestions put forward by the reviewer **2efs**. In the research direction of SFT data selection, neither our method nor some previous works have paid attention to the issue of data overlap between the training set and the test set. We agree with your point of view and will focus on the data overlap issue in our subsequent research.
>
> Regarding the issue of token overlap you mentioned, we believe that it is inappropriate to count the overlap just between tokens. Because individual tokens don't have any practical meaning on their own. We calculated the n-gram overlap degree (n = 15) between the data subsets screened by different methods and the test data of different tasks. If the same n-gram tokens appear between two pieces of data, we consider this part of the data to be the same. The results are shown as follows, the numbers in the table represent the proportion of token-overlapping samples in each test set, (overlap samples/total test samples). We found that the proportion of overlapping samples between the training set and the test set is very low. Therefore, the impact of overlapping samples on the test tasks is actually very weak.
>
> ||ifeval|gsm|mmlu|code|bbh|
> |-|-|-|-|-|-|
> |ifd|0|0|7.15E-04|1.83E-02|0|
> |less|0|0|2.50E-03|2.44E-02|1.90E-03|
> |select|0|7.60E-04|2.80E-03|5.49E-02|1.20E-03|
> |entropy|0|0|3.58E-04|6.71E-02|0|
> |diverse|0|0|7.00E-05|2.44E-02|0|
> |zip|0|0|3.58E-04|2.44E-02|2.90E-03|
> |random|0|0|7.15E-04|4.88E-02|0|
> |length|0|0|4.29E-04|6.09E-02|0|
>
> **We have already submitted the revised paper and made corresponding modifications in response to the comments of all the reviewers.**

---

> > ### Comment · Reviewer_2efs · 2024-12-03
> >
> > Thank you for checking the overlap. I am satisfied with an answer and keep my rating.

---

### Meta-Review · Area_Chair_ic8b · 2024-12-19

**Metareview:**

The paper analyzes methods for selecting a high quality dataset for LLM supervised fine tuning (SFT). This task is highly motivated given that many papers show the crucial role of data quality in LLM training. An additional strength mentioned by the reviewers is the insights provided in the paper, showing that many of the existing methods for data selection are in fact no better than random selection.

The authors provide additional insights and some supporting experiments, but according to the reviews, these require additional and deeper analysis to meet the bar of ICLR. The authors provided some clarifications and additional data in the rebuttal stage, but the reviews did not find it to be sufficient, and this key issue remains. Other than the need for a deeper analysis, some concerns were raised regarding the novelty compared to previous works (yhuK, MLme). This issue might be mitigated given a deeper analysis addressing the points raised in the reviews.

Concluding, the paper deals with an important problem and has potential, but it requires more work.

**Additional Comments On Reviewer Discussion:**

see meta review

---

### Decision · Program_Chairs · 2025-01-22

Reject